# On-the-fly Operation Batching
# in Dynamic Computation Graphs

**Graham Neubig**[*]
Language Technologies Institute
Carnegie Mellon University
gneubig@cs.cmu.edu

**Yoav Goldberg**[*]
Computer Science Department
Bar-Ilan University
yogo@cs.biu.ac.il

**Chris Dyer**
DeepMind
cdyer@google.com

## Abstract

Dynamic neural network toolkits such as PyTorch, DyNet, and Chainer offer more flexibility for implementing models that cope with data of varying dimensions and structure, relative to toolkits that operate on statically declared computations (e.g., TensorFlow, CNTK, and Theano). However, existing toolkits—both static and dynamic—require that the developer organize the computations into the batches necessary for exploiting high-performance algorithms and hardware. This batching task is generally difficult, but it becomes a major hurdle as architectures become complex. In this paper, we present an algorithm, and its implementation in the DyNet toolkit, for automatically batching operations. Developers simply write minibatch computations as aggregations of single instance computations, and the batching algorithm seamlessly executes them, on the fly, using computationally efficient batched operations. On a variety of tasks, we obtain throughput similar to that obtained with manual batches, as well as comparable speedups over single-instance learning on architectures that are impractical to batch manually.[2]

## 1 Introduction

Modern CPUs and GPUs evaluate batches of arithmetic operations significantly faster than the sequential evaluation of the same operations. For example, performing elementwise operations takes nearly the same amount of time on the GPU whether operating on tens or on thousands of elements, and multiplying a few hundred different vectors by the same matrix is significantly slower than executing a single (equivalent) matrix–matrix product using an optimized GEMM implementation on either a GPU or a CPU. Thus, careful grouping of operations into batches that can execute efficiently in parallel is crucial for making the most of available hardware resources.

Today, developers who write code to train neural networks are responsible for crafting most of this batch handling by hand. In some cases this is easy: when inputs and outputs are naturally represented as fixed sized tensors (e.g., images of a fixed size such those in the MNIST and CIFAR datasets, or regression problems on fixed sized vector inputs), and the computations required to process each instance are instance-invariant and expressible as standard operations on tensors (e.g., a series of matrix multiplications, convolutions, and elementwise nonlinearities), a suitably flexible tensor library

---

[*]Authors contributed equally.

[2]The proposed algorithm is implemented in DyNet (`http://dynet.io/`), and can be activated by using the "`--dynet-autobatch 1`" command line flag.

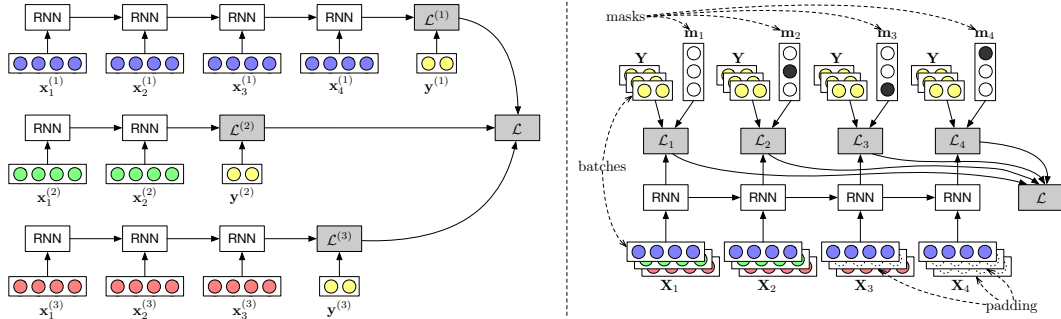

Figure 1: Two computation graphs for computing the loss on a minibatch of three training instances consisting of a sequence of input vectors paired with a fixed sized output vector. On the left is a "conceptual" computation graph which shows the operations associated with computing the losses individually for each sequence and then aggregating them. The same computation is executed by the right-hand ("batched") computation graph: it aggregates the inputs in order to make better use of modern processors. This comes with a price in complexity—the variable length of the sequences requires padding and masking operations. Our aim is for the user to specify the conceptual computation on the left, and let the framework take care of its efficient execution.

that provides efficient implementations of higher-order generalizations of low-order operations makes manual batching straightforward. For example, by adding a leading or trailing dimension to the tensors representing inputs and outputs, multiple instances can be straightforwardly represented in a single data structure. In other words: in this scenario, the developer conceives of and writes code for the computation on an individual instance, packs several instances into a tensor as a "minibatch", and the library handles executing these efficiently in parallel.

Unfortunately, this idealized scenario breaks when working with more complex architectures. Deep learning is increasingly being applied to problems whose inputs, outputs and intermediate representations do not fit easily into fixed sized tensors. For example, images vary in size and sequences in length; data may be structured as trees [29] or graphs [4, 17, 27], or the model may select its own computation conditional on the input [16, 28, 33]. In all these cases, while the desired computation is easy enough to write for a single instance, organizing the computational operations so that they make optimally efficient use of the hardware is nontrivial. Indeed, many papers that operate on data structures more complicated than sequences have avoided batching entirely [8, 15, 25]. In fact, until last year [7, 20], *all* published work on recursive (i.e., tree-structured) neural networks appears to have used single instance training.

The premise of this work is that operation batching should not be the responsibility of the user, but instead should be a service provided by the framework. The user should only be responsible for specifying a large enough computation so that batching is possible (i.e, summing the losses of several instances, such as one sees in the left side of Figure 1), and the framework should take care of the lower-level details of operation batching, much like optimizing compilers or JIT optimizers in interpreted languages do.[3]

We take a large step towards this goal by introducing an efficient algorithm—and a corresponding implementation—for automatic batching in dynamically declared computation graphs.[4] Our method relies on separating the graph construction from its execution, using operator overloading and lazy

evaluation (§2). Once this separation is in place, we propose a fast batching heuristic that can be performed in real time, for each training instance (or minibatch), between the graph construction and its execution (§3). We extend the DyNet toolkit [21] with this capability. From the end-user's perspective, the result is a simple mechanism for exploiting efficient data-parallel algorithms in networks that would be cumbersome to batch by hand. The user simply defines the computation independently for each instance in the batch (using standard Python or C++ language constructs), and the framework takes care of the rest. Experiments show that our algorithm compares favorably to manually batched code, that significant speed improvements are possible on architectures with no straightforward manual batching design, and that we obtain better performance than TensorFlow Fold [19], an alternative framework built to simulate dynamic graph definition and automatic batching on top of TensorFlow (§4).

## 2 Batching: Conception vs. Efficient Implementation

To illustrate the challenges with batching, consider the problem of predicting a real-valued vector conditional on a sequence of input vectors (this example is chosen for its simplicity; experiments are conducted on more standard tasks). We assume that an input sequence of vectors is read sequentially by an RNN, and then the final state is used to make a prediction; the training loss is the Euclidean distance between the prediction and target. We compare two algorithms for computing this code: a naïve, but developer-friendly one (whose computation graph is shown in the left part of Figure 1), which reflects how one conceives of what a batch loss computation is; and a computationally efficient— but more conceptually complex—version that batches up the computations so they are executed in parallel across the sequences (the right part of Figure 1).

**Naïve (developer-friendly) batched implementation**    The left part of Figure 1 shows the computations that must be executed to compute the losses associated with three ($b = 3$) training instances, implemented naïvely. Pseudo-code for constructing the graph for each of the RNNs on the left using a dynamic declaration framework is as follows:

> **function** RNN-REGRESSION-LOSS($\mathbf{x}_{1:n}, \mathbf{y}; (\mathbf{W}, \mathbf{U}, \mathbf{b}, \mathbf{c}) = \boldsymbol{\theta}$)
>> $\mathbf{h}_0 = \mathbf{0}$                                                    ▷ Initial state of the RNN; $\mathbf{h}_t \in \mathbb{R}^d$.
>> **for** $t \in 1, 2, \ldots, n$ **do**
>>> $\mathbf{h}_t = \tanh(\mathbf{W}[\mathbf{h}_{t-1}; \mathbf{x}_t] + \mathbf{b})$
>>
>> $\hat{\mathbf{y}} = \mathbf{U}\mathbf{h}_n + \mathbf{c}$
>> $\mathcal{L} = ||\hat{\mathbf{y}} - \mathbf{y}||_2^2$
>> **return** $\mathcal{L}$

Note that the code does not compute any value, but constructs a symbolic graph describing the computation. This can then be integrated into a batched training procedure:

> **function** TRAIN-BATCH-NAIVE($\mathcal{T} = \{(\mathbf{x}_{1:n^{(i)}}^{(i)}, \mathbf{y}^{(i)})\}_{i=1}^b; \boldsymbol{\theta}$)
>> NEW-GRAPH()
>> **for** $i \in 1, 2, \ldots, b$ **do**                                   ▷ Naïvely loop over elements of batch.
>>> $\mathcal{L}^{(i)} = $ RNN-REGRESSION-LOSS($\mathbf{x}_{1:n^{(i)}}^{(i)}, \mathbf{y}^{(i)}; \boldsymbol{\theta}$)      ▷ Single instance loss.
>>
>> $\mathcal{L} = \sum_i \mathcal{L}^{(i)}$                               ▷ Aggregate losses for all elements in batch.
>> FORWARD($\mathcal{L}$)
>> $\frac{\partial \mathcal{L}}{\partial \boldsymbol{\theta}} = $ BACKWARD($\mathcal{L}$)
>> $\boldsymbol{\theta} = \boldsymbol{\theta} - \eta \frac{\partial \mathcal{L}}{\partial \boldsymbol{\theta}}$

This code is simple to understand, uses basic flow control present in any programming language and simple mathematical operations. Unfortunately, executing it will generally be quite inefficient, since in the resulting computation graph each operation is performed sequentially without exploiting the fact that similar operations are being performed across the training instances.

**Efficient manually batched implementation**    To make good use of efficient data-parallel algorithms and hardware, it is necessary to batch up the operations so that the sequences are processed in parallel. The standard way to achieve this is by aggregating the inputs and outputs, altering the code as follows:

**function** RNN-REGRESSION-BATCH-LOSS($\mathbf{X}_{1:n_{\max}}, \mathbf{Y}, n^{(1:b)}; (\mathbf{W}, \mathbf{U}, \mathbf{b}, \mathbf{c}) = \boldsymbol{\theta}$)

$\quad \mathbf{M} = \mathbf{0}$ $\qquad\qquad\qquad\qquad\qquad\qquad\qquad$ ▷ Build loss mask; $\mathbf{M} \in \mathbb{R}^{b \times n_{\max}}$.

$\quad$ **for** $i \in 1, 2, \ldots, b$ **do**

$\qquad \mathbf{M}_{[i, n^{(i)}]} = 1$ $\qquad\qquad\qquad$ ▷ Position where the final symbol in sequence $i$ occurs.

$\quad \mathbf{H}_0 = \mathbf{0}$ $\qquad\qquad\qquad$ ▷ Initial states of the RNN (one per instance); $\mathbf{H}_t \in \mathbb{R}^{d \times b}$.

$\quad$ **for** $t \in 1, 2, \ldots, n_{\max}$ **do**

$\qquad \mathbf{H}_t = \tanh(\mathbf{W}[\mathbf{H}_{t-1}; \mathbf{X}_t] + \mathbf{b})$ $\qquad\qquad$ ▷ Addition broadcasts $\mathbf{b}$ over columns.

$\qquad \hat{\mathbf{Y}}_t = \mathbf{U}\mathbf{H}_t + \mathbf{c}$ $\qquad\qquad\qquad$ ▷ Addition broadcasts $\mathbf{c}$ over columns.

$\qquad \mathcal{L}_t = ||(\hat{\mathbf{Y}}_t - \mathbf{Y})(\mathbf{m}_t \mathbf{1}^\top)||_{\mathcal{F}}^2$ $\qquad$ ▷ Compute masked losses ($\mathbf{m}_t$ is the $t$th column of $\mathbf{M}$).

$\quad \mathcal{L} = \sum_t \mathcal{L}_t$

$\quad$ **return** $\mathcal{L}$

**function** TRAIN-BATCH-MANUAL($\mathcal{T} = \{(\mathbf{x}_{1:n^{(i)}}^{(i)}, \mathbf{y}^{(i)})\}_{i=1}^b; \boldsymbol{\theta}$)

$\quad n_{\max} = \max_i n^{(i)}$

$\quad$ **for** $t \in 1, 2, \ldots, n_{\max}$ **do** $\qquad\qquad\qquad$ ▷ Build sequence of batch input matrices.

$\qquad \mathbf{X}_t = \mathbf{0} \in \mathbb{R}^{d \times b}$

$\qquad$ **for** $i \in 1, 2, \ldots, b$ **do**

$\qquad\qquad \mathbf{X}_{t,[\cdot,i]} = \mathbf{x}_t^{(i)}$ **if** $t \leq n^{(i)}$ **otherwise** 0 $\qquad\qquad$ ▷ The $i$th column of $\mathbf{X}_t$.

$\quad \mathbf{Y} = [\mathbf{y}^{(1)}\ \mathbf{y}^{(2)}\ \cdots\ \mathbf{y}^{(b)}]$ $\qquad\qquad\qquad$ ▷ Build batch of output targets.

$\quad$ NEW-GRAPH() $\quad$ ▷ Now that inputs are constructed, create graph, evaluate loss and gradient.

$\quad \mathcal{L} = $ RNN-REGRESSION-BATCH-LOSS($\mathbf{X}_{1:n_{\max}}, \mathbf{Y}, n^{(1:b)}; \boldsymbol{\theta}$)

$\quad$ FORWARD($\mathcal{L}$)

$\quad \frac{\partial \mathcal{L}}{\partial \boldsymbol{\theta}} = $ BACKWARD($\mathcal{L}$)

$\quad \boldsymbol{\theta} = \boldsymbol{\theta} - \eta \frac{\partial \mathcal{L}}{\partial \boldsymbol{\theta}}$

This code computes the same value as the naïve implementation, it does so more efficiently, and it is significantly more complicated. Because the sequences processed by RNNs will generally be of different lengths (which is precisely why RNNs are useful!), it is necessary to pad the input representation with dummy values, and also to mask out the resulting losses at the right times. While these techniques are part of the inventory of skills that a good ML engineer has, they increase the difficulty of implementation and probability that bugs will be present in the code.

**Implementation comparison** The naïve algorithm has two advantages over manual batching. First, it is easy to implement: the way we conceive of a model is the way it is implemented, and errors with padding, masking, and batching are avoided. Second, the naïve algorithm aggregates *any* single instance loss, whereas manual batching efforts are generally problem specific. For these reasons, one should strongly prefer the first algorithm; however, for efficiency reasons, batching matters. In the next section we turn to the problem of how to efficiently execute naïve computation graphs so that they can take advantage of efficient batched implementations of operations. This provides the best of both worlds to developers: code is easy to write, but execution is fast.

## 3 An Algorithm for On-the-fly Batching

Manual batching, discussed in the previous section, mostly operates by *aggregating input instances* and feeding them through a network. In RNNs, this means aggregating inputs that share a time step. This often require padding and masking, as input sizes may differ. It also restricts the kinds of operations that can be batched. In contrast, our method *identifies and aggregates computation graph nodes* that can be executed in a batched fashion for a given graph. This reduces the need for workarounds such as padding and masking, allows for seamless efficient execution also in architectures which are hard to conceptualize in the input-centric paradigm, and allows for the identification of batching opportunities that may not be apparent from an input-centric view.

Our batching procedure operates in three steps (1) graph definition, (2) operation batching, and (3) computation. Here, steps (1) and (3) are shared with standard execution of computation graphs, while (2) corresponds to our proposed method.

### 3.1 Graph Definition

First, we define the graph that represents the computation that we want to perform. From the user's perspective, this is done by simply performing computation that they are interested in performing, such as that defined in the RNN-REGRESSION-LOSS function from the previous example. While it is common for dynamic graph frameworks to interleave the graph definition and its forward execution,

we separate these parts by using *lazy evaluation*: we only perform forward evaluation when a resulting value is requested by the user through the calling of the FORWARD function. The graph can be further extended after a call to FORWARD, and further calls will lazily evaluate the delta of the computation. This allows the accumulation of large graph chunks before executing forward computations, providing ample opportunities for operation batching.

## 3.2  Operation Batching

Next, given a computation graph, such as the one on the left side of Figure 1, our proposed algorithm converts it into a graph where operations that can be executed together are batched together. This is done in the two step process described below.

**Computing compatibility groups**   We first partition the nodes into compatibility groups, where nodes in the same group have the potential for batching. This is done by associating each node with a signature such that nodes that share the same signature are guaranteed to be able to be executed in a single operation if their inputs are ready. Signatures vary depending on the operation the node represents. For example, in nodes representing element-wise operations, all nodes with the same operation can be batched together, so the signature is simply the operation name (`tanh`, `log`, ...). In nodes where dimensions or other information is also relevant to whether the operations can be batched, this information is also included in the signature. For example, a node that picks a slice of the input matrix will also be dependent on the matrix size and range to slice, so the signature will look something like `slice-400x500-100:200`. In some other cases (e.g. a parameterized matrix multiply) we may remember the specific node ID of one of the inputs (e.g. `node123` representing the matrix multiply parameters) while generalizing across other inputs (e.g. data or hidden state vectors on the right-hand side), resulting in a signature that would look something like `matmul-node123-400x1`. A more thorough discussion is given in Appendix A.

**Determining execution order**   A computation graph is essentially a job dependency graph where each node depends on its input (and by proxy the input of other preceding nodes on the path to its inputs). Our goal is to select an execution order in which (1) each node is executed after its dependencies; and (2) nodes that have the same signature and do not depend on each other are scheduled for execution on the same step (and will be executed in a single batched operation). Finding an optimal execution order that maximizes the amount of batching in the general case is NP hard [24]. We discuss two heuristic strategies for identifying execution orders that satisfy these requirements.

*Depth-based batching* is used as a method for automatic batching in TensorFlow Fold [19]. This is done by calculating the depth of each node in the original computation graph, defined as the maximum length from a leaf node to the node itself, and batching together nodes that have an identical depth and signature. By construction, nodes of the same depth are not dependent on each-other, as all nodes will have a higher depth than their input, and thus this batching strategy is guaranteed to satisfy condition (1) above. However, this strategy will also miss some good batching opportunities. For example, the loss function calculations in Figure 1 are of different depths due to the different-lengthed sequences, and similar problems will occur in recurrent neural network language models, tree-structured neural networks, and a myriad of other situations.

*Agenda-based batching* is a method we propose that does not depend solely on depth. The core of this method is an agenda that tracks "available" nodes that have no unresolved dependencies. For each node, a count of its unresolved dependencies is maintained; this is initialized to be the number of inputs to the node. The agenda is initialized by adding nodes that have no incoming inputs (and thus no unresolved dependencies). At each iteration, we select a node from the agenda together with all of the available nodes in the same signature, and group them into a single batch operation. These nodes are then removed from the agenda, and the dependency counter of all of their successors are decremented. Any new zero-dependency nodes are added to the agenda. This process is repeated until all nodes have been processed.

How do we prioritize between multiple available nodes in the agenda? Intuitively, we want to avoid prematurely executing nodes if there is a potential for more nodes of the same signature to be added to the agenda at a later point, resulting in better batching. A good example of this from our running example in Figure 1 is the loss-calculating nodes, which will be added to the agenda at different points due to becoming calculable after different numbers of RNN time steps. To capture this intuition, we introduce a heuristic method for prioritizing nodes based on the *average depth* of all nodes with their

signature, such that nodes with a lower average depth will be executed earlier. In general (with some exceptions), this tends to prioritize nodes that occur in earlier parts of the graph, which will result in the nodes in the later parts of the graph, such as these loss calculations, being executed later and hopefully batched together.[5]

Finally, this non-trivial batching procedure must be executed quickly so that overhead due to batch scheduling calculations doesn't cancel out the efficiency gains from operation batching. To ensure this, we perform a number of optimizations in the implementation, which we detail in Appendix B.

### 3.3 Forward-backward Graph Execution and Update

Once we have determined an execution order (including batching decisions), we perform calculations of the values themselves. In standard computation graphs, forward computation is done in topological order to calculate the function itself, and backward calculation is done in reverse topological order to calculate gradients. In our automatically batched evaluation, the calculation is largely similar with two exceptions:

**Single→batch node conversion**  First, it is necessary to convert single nodes into a batched node, which also requires modification of the underlying operations such as converting multiple matrix-vector operations $\mathbf{W}\mathbf{h}_i$ to a single matrix-matrix operation $\mathbf{W}\mathbf{H}$. This is done internally in the library, while the user-facing API maintains the original unbatched computation graph structure, making this process invisible to the user.

**Ensuring contiguous memory**  To ensure that operations can be executed as a batch, the inputs to the operations (e.g. the various vectors $\mathbf{h}_t^{(i)}$) must be arranged in contiguous memory (e.g. a matrix $\mathbf{H}_t$). In some cases, it is necessary to perform a memory copy to arrange these inputs into contiguous memory, but in other cases the inputs are already contiguous and in the correct order, and in these cases we can omit the memory copy and use the inputs as-is.[6]

## 4  Experiments

In this section we describe our experiments, designed to answer three main questions: (1) in situations where manual batching is easy, how close can the proposed method approach the efficiency of a program that uses hand-crafted manual batching, and how do the depth-based and agenda-based approaches compare (§4.1)? (2) in situations where manual batching is less easy, is the proposed method capable of obtaining significant improvements in efficiency (§4.2)? (3) how does the proposed method compare to TensorFlow Fold, an existing method for batching variably structured networks within a static declaration framework (§4.3)?

### 4.1 Synthetic Experiments

Our first experiments stress-test our proposed algorithm in an ideal case for manual batching. Specifically, we train a model on a bi-directional LSTM sequence labeler [12, 23], on synthetic data where every sequence to be labeled is the same length (40). Because of this, manual batching is easy because we don't have to do any padding or adjustment for sentences of different lengths. The network takes as input a size 200 embedding vector from a vocabulary of size 1000, has 2 layers of 256 hidden node LSTMs in either direction, then predicts a label from one of 300 classes. The batch size is 64.[7]

Within this setting we test various batching settings: Without or with manual mini-batching where we explicitly batch the word vector lookup, LSTM update, and loss calculation for each time step.

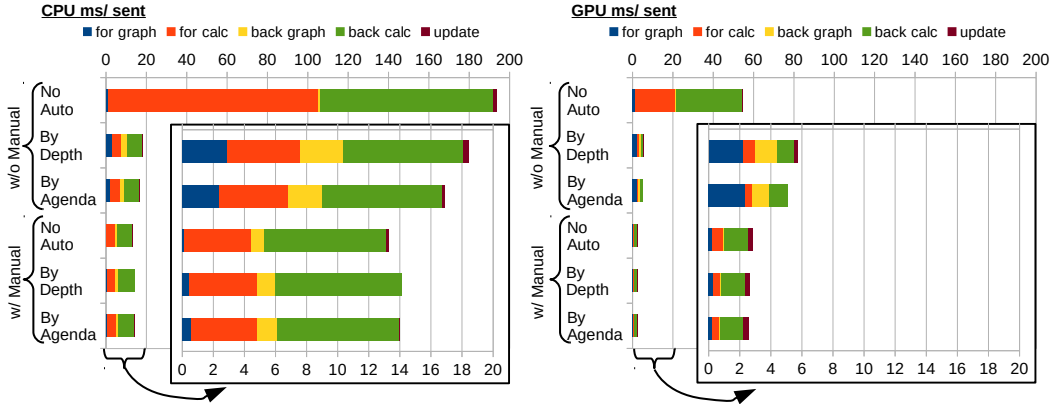

Figure 2: Computation time for forward/backward graph construction or computation, as well as parameter update for a BiLSTM tagger without or with manual batching, and without, with depth-based, or with agenda-based automatic batching.

Without on-the-fly batching (NOAUTO), with depth-based autobatching (BYDEPTH), or with agenda-based autobatching (BYAGENDA). We measure the speed of each method by ms/sec and also break down the percentage of computation time spent in (1) forward graph creation/on-the-fly batching, (2) forward computation, (3) backward graph creation, (4) backward computation, (5) parameter update.

The results can be found in Figure 2. First, comparing the first row with the second two, we can see that the proposed on-the-fly batching strategy drastically reduces computation time per sentence, with BYAGENDA reducing per-sentence computation time from 193ms to 16.9ms on CPU and 54.6ms to 5.03ms on GPU, resulting in an approximately 11-fold increase in sentences processed per second (5.17→59.3 on CPU and 18.3→198 on GPU). BYAGENDA is faster than BYDEPTH by about 15–30%, demonstrating that our more sophisticated agenda-based strategy is indeed more effective at batching together operations.

Next, compared to manual batching without automatic batching (the fourth row), we can see that fully automatic batching with no manual batching is competitive, but slightly slower. The speed decrease is attributed to the increased overhead for computation graph construction and batch scheduling. However, even in this extremely idealized scenario where manual batching will be most competitive, the difference is relatively small ($1.27\times$ on CPU and $1.76\times$ on GPU) compared to the extreme difference between the case of using no batching at all. Given that automatic batching has other major advantages such as ease of implementation, it may be an attractive alternative even in situations where manual batching is relatively easy.

Finally, if we compare the fourth and fifth/sixth rows, we can see that on GPU, even with manual batching, automatic batching still provides gains in computational efficiency, processing sentences up to 1.1 times faster than without automatic batching. The reason for this can be attributed to the fact that our BiLSTM implementation performs manual batching across sentences, but not across time steps within the sentence. In contrast, the auto-batching procedure was able to batch the word embedding lookup and softmax operations across time-steps as well, reducing the number of GPU calls and increasing speed. This was not the case for CPU, as there is less to be gained from batching these less expensive operations.

## 4.2   Experiments on Difficult-to-batch Tasks

Next, we extend our experiments to cases that are increasingly more difficult to manually batch. We use realistic dimension sizes for the corresponding tasks, and batches of size $b = 64$. Exact dimensions and further details on training settings are in Appendix C.

**BiLSTM:** This is similar to the ideal case in the previous section, but trained on actual variable length sequences.

**BiLSTM w/char:** This is the same as the BiLSTM tagger above, except that we use an additional BiLSTM over characters to calculate the embeddings over rare words. These sorts of

Table 1: Sentences/second on various training tasks for increasingly challenging batching scenarios.

| Task | CPU | | | GPU | | |
|---|---|---|---|---|---|---|
| | NOAUTO | BYDEPTH | BYAGENDA | NOAUTO | BYDEPTH | BYAGENDA |
| BiLSTM | 16.8 | 139 | **156** | 56.2 | 337 | **367** |
| BiLSTM w/ char | 15.7 | 93.8 | **132** | 43.2 | 183 | **275** |
| TreeLSTM | 50.2 | 348 | **357** | 76.5 | **672** | 661 |
| Transition-Parsing | 16.8 | 61.0 | **61.2** | 33.0 | 89.5 | **90.1** |

character-based embeddings have been shown to allow the model to generalize better [18], but also makes batching operations more difficult, as we now have a variably-lengthed encoding step that may or may not occur for each of the words in the input.

**Tree-structured LSTMs:** This is the Tree-LSTM model of [31]. Here, each instance is a tree rather than a sequence, and the network structure follows the tree structures. As discussed in the introduction, this architecture is notoriously hard to manually batch.

**Transition-based Dependency Parsing:** The most challenging case we evaluate is that of a transition-based system, such as a transition based parser with LSTM-based feature-extraction [8, 9, 13] and exploration-based training [2, 5, 10]. Here, a sequence is encoded using an LSTM (or a bi-LSTM), followed by a series of predictions. Each prediction based on a subset of the encoded vectors, and the vectors that participate in each prediction, as well as the loss, are determined by the outcomes of the previous predictions. Here, batching is harder yet as the nature of the computation interleaves sampling from the model and training, and requires calling FORWARD at each step, leaving the automatic-batcher very little room to play with. However, with only a small change to the computation, we can run $b$ different parsers "in parallel", and potentially share the computation across the different systems in a given time-step. Concretely, we use a modified version of the BIST parser [14].

From the results in Table 1, we can see that in all cases automatic batching gives healthy improvements in computation time, 3.6x–9.2× on the CPU, and 2.7–8.6× on GPU. Furthermore, the agenda-based heuristic is generally more effective than the depth-based one.

### 4.3 Comparison to TensorFlow Fold

We compare the TensorFlow Fold reference implementation of the Stanford Sentiment Treebank regression task [30], using the same TreeLSTM architecture [31]. Figure 3 shows how many trees are processed per second by TF (excluding both evaluation of the dev set and static graph construction/optimization) on GPU and CPU relative to the performance of the BYAGENDA algorithm in DyNet (including graph construction time). The DyNet performance is better across the board stratified by hardware type. Furthermore, DyNet has greater throughput on CPU than TensorFlow Fold on GPU until batch sizes exceed 64. Additionally, we find that with single instance training, DyNet's sequential evaluation processes 46.7 trees/second on CPU, whereas autobatching processes 93.6 trees/second. This demonstrates that in complex

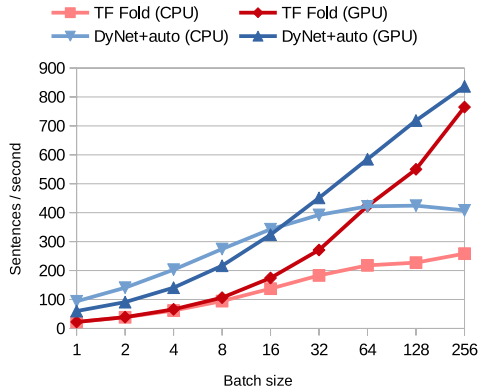

Figure 3: Comparison of runtime performance between TensorFlow Fold and DyNet with autobatching on TreeLSTMs (trees/sec).

architectures like TreeLSTMs, there are opportunities to batch up operations inside a single training instance, which are exploited by our batching algorithm. In addition, it should be noted that the DyNet implementation has the advantage that it is much more straightforward, relying on simple Python data structures and flow control to represent and traverse the trees, while the Fold implementation requires implementing the traversal and composition logic in a domain specific functional programming language (described in Section 3 of Looks et al. [19]).

# 5   Related Work

Optimization of static algorithms is widely studied, and plays an important role in numerical libraries used in machine learning. Our work is rather different since the code/workload (as represented by the computation graph) is dynamically specified and must be executed rapidly, which precludes sophisticated statistic analysis. However, we review some of the important related work here.

Automatic graph optimization and selection of kernels for static computation graphs is used in a variety of toolkits, including TensorFlow [1] and Theano [6]. Dynamic creation of optimally sized minibatches (similar to our strategy, except the computation graph is assumed to be static) that make good use of hardware resources has also been proposed for optimizing convolutional architectures [11]. The static nature of the computation makes this tools closer to optimizing compilers rather than efficient interpreters which are required to cope with the dynamic workloads encountered when dealing with dynamically structured computations.

Related to this is the general technique of automatic vectorization, which is a mainstay of optimizing compilers. Recent work has begun to explore vectorization in the context of interpreted code which may cannot be compiled [26]. Our autobatching variant of DyNet similarly provides vectorized primitives that can be selected dynamically.

Further afield, the problem of scheduling with batching decisions has been widely studied in operations research since at least the 1950s (for a recent survey, see [24]). Although the OR work deals with similar problems (e.g., scheduling work on machines that can process a 'family' of related item with minimal marginal cost over a single item), the standard algorithms from this field (which are often based on polynomial-time dynamic programs or approximations to NP-hard search problems) are too computationally demanding to execute in the inner loop of a learning algorithm.

# 6   Conclusion

Deep learning research relies on empirical exploration of architectures. The rapid pace of innovation we have seen in the last several years has been enabled largely by tools that have automated the error-prone aspects of engineering, such as writing code that computes gradients. However, our contention is that operation batching is increasingly becoming another aspect of model coding that is error prone and amenable to automation.

Our solution is a framework that lets programmers express computations naturally and relies on a smart yet lightweight interpreter to figure out how to execute the operations efficiently. Our hope is that this will facilitate the creation of new classes of models that better cope with the complexities of real-world data.

**Acknowledgements:**  The work of YG is supported by the Israeli Science Foundation (grant number 1555/15) and by the Intel Collaborative Research Institute for Computational Intelligence (ICRI-CI).

## Footnotes

[3] This is in contrast to other existing options for automatic batching such as TensorFlow Fold, which require the user to learn an additional domain-specific language to turn computation into a format conducive to automatic batching [19].

[4] Computation graphs (often represented in a form called a Wengert list) are the data structures used to structure the evaluation of expressions and use reverse mode automatic differentiation to compute their derivatives [3]. Broadly, learning frameworks use two strategies to construct these: static and dynamic. In static toolkits (e.g., Theano [6], Tensorflow [1]) the computation graph is defined once and compiled, and then examples are fed into the same graph. In contrast, dynamic toolkits (e.g., DyNet [21], Chainer [32], PyTorch [http://pytorch.org]) construct the computation graph for each training instance (or minibatch) as the forward computation is executed. While dynamic declaration means that each minibatch can have its own computational architecture, the user is still responsible for batching operations themselves.

[5]Even given this prioritization method it is still possible to have ties, in which case we break ties by calculating "cheap" operations (e.g. `tanh` and other elementwise ops) before "heavy" ones (e.g. matrix multiplies).

[6]The implication of this is that batched computation will take up to twice as much memory as unbatched computation, but in practice the memory usage is much less than this. Like manually batched computation, memory usage can be controlled by adjusting the batch size appropriately so it fits in memory.

[7]Experiments were run on a single Tesla K80 GPU or Intel Xeon 2.30GHz E5-2686v4 CPU. To control for variance in execution time, we perform three runs and report the fastest. We do not report accuracy numbers, as the functions calculated and thus accuracies are the same regardless of batching strategy.

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
