[Supplementary Material]

## A  Details on Node Signatures

Each node has a "signature," such that nodes with identical signatures can be batched together. With few exceptions, nodes can only be batched together if they perform the same operation, so the identity of the operation the node performs will be a necessary part of the signature. In addition, there may be additional constraints on what nodes can be batched together based on the nature of the operation to be performed. We demonstrate the signatures for a few illustrative classes of operations below:

**Component-wise operations** such as "tanh" or "log" will perform exactly the same work regardless of the shape of the tensors involved. For these simple operations, the signature is simply the identity of the operation (e.g. `tanh` or `log`) with no additional constraints. This is also true for component-wise operations that take multiple arguments such as sums or component-wise multiplications, as long as they do not involve broadcasting, which will be discussed in the following items.

**Dimension-sensitive operations** require additional restrictions. For example, matrix multiplies can generally only be performed on inputs where the dimensions match, so if we have several $\mathbf{W}_i \mathbf{h}_i$ operations we will only be able to batch them together if $\mathbf{W}_i$ and $\mathbf{h}_i$ are the same dimension across all elements $i$. In these cases, we explicitly specify the necessary dimensions in the signature (e.g. `mult-256×256-256` if $\mathbf{W}_i$ was a 256×256 matrix and $\mathbf{b}_i$ was a length-256 vector), preventing inputs with incompatible dimensions from being processed together.

**Operations with shared elements** such as a matrix–vector multiply $\mathbf{W}\mathbf{h}_i$ where same matrix is applied to all the vectors, are both common and the source of most potential gains from operation batching. The reason why these operations are important is because we can perform explicit optimizations such as concatenating all of the $\mathbf{h}_i$ vectors into a matrix $\mathbf{H}$ and performing a single matrix–matrix multiplication $\mathbf{W}\mathbf{H}$. To take advantage of this, if $\mathbf{W}$ is represented as node $n_{\mathbf{W}}$, we can define a signature `mult-`$n_{\mathbf{W}}$`-256`, where operations that share their left side but may have different right sides are grouped together. Matrix multiplication can have either a shared or un-shared signature.

**Unbatchable operations** are operations that either cannot be batched together trivially, or would not benefit significantly from batching.

One thing that should be noted is that for some nodes, like the matrix multiplies $\mathbf{A}\mathbf{x}$ in the example or affine transforms $\mathbf{A}\mathbf{x} + \mathbf{y}$, which signature to use is not clear. If some of the elements are shared parameters, it would be preferable to use a signature that shares these parameters to take advantage of efficient implementations such as the one mentioned above. However, if all of the elements of the multiply or affine transform are unique, then it is better to use the simpler dimension-sensitive operations.

In our implementation, we use a simple heuristic: because multiplies and affine transforms in neural networks tend to have the parameters in the positions of $\mathbf{A}$ and $\mathbf{y}$, and the elements in the $\mathbf{x}$ position tend to be input-dependent, we use signatures that share the elements in the $\mathbf{A}$ and $\mathbf{y}$ positions but do not share the elements in the $\mathbf{x}$ position.

## B  Optimizations for Fast Graph Calculation

In order to ensure that the increased complexity of automatic batching does not introduce unacceptable overhead in our calculation, we took care to efficiently implement the different parts of the algorithm using sophisticated but fairly standard optimization techniques. These include:

- Minimizing the number of memory allocations and preferring stack allocation of fixed-size memory to heap allocation of variable-sized memory.
- Implementing specialized linked-list-style data structures in contiguous memory to avoid expensive-to-construct vectors of vectors.
- Computing node signatures as integer hash values instead of strings.
- Implementing optimized GPU kernels to perform sparse-to-dense and dense-to-sparse memory copies for use when copying operations results to/from contiguous memory for use in batched nodes.

Details of all of these optimizations can be found in the open source implementation in DyNet.[8]

## C   Experimental Settings

The first three experiments are based on implementations in the DyNet benchmark repository,[9].

**BiLSTM (`bilstm-tagger-bulk`):** As our tagging data, we use data from the named entity recognition task Models were trained and tested on the WikiNER English Corpus [22], and all words with frequency less than five were treated as unknowns. The network was single-layer with word embeddings and LSTMs in either direction containing 256 nodes each.

**BiLSTM w/char (`bilstm-tagger-withchar-bulk`):** The settings for the BiLSTM tagger with character embeddings are the same as above, but with the addition of character-based LSTMs for unknown words. The character embeddings are of size 64, and the character LSTMs are 128 in both directions.

**Tree-structured LSTMs (`treenn-bulk`):** Tree LSTMs are trained on the Stanford Sentiment Treebank regression task [30]. These similarly use word embedding and node sizes of 256. The models are trained to predict the labels at the each node in the tree.

For our final experiment, we modified a version of the publicly available transition-based version (`barchybrid`) of the BISTPARSER[10][14]. Our modified code is available in the DyNet benchmark repository.

**Transition-based Dependency Parsing:** The parser was modified to perform aggregate batching by running several parsers in-parallel and aggregating decisions in a given time-step across the different parsers. In contrast to the other benchmarks in this paper which are implemented in C++, this is a python-based implementation. We measure the training time of one iteration over the training set of the publicly available English Universal Dependencies Treebank,[11] containing 12K sentences. We use the default settings of the parser (100 dim word embeddings, 25 dim POS embeddings, 25 dim relation embeddings, 200 dim LSTM layers, and a 100 dim hidden layer in the prediction MLP), as well as the flags `--userlmost --userl --bibi-lstm`.

## Footnotes

[8]`http://dynet.io/`

[9]`https://github.com/neulab/dynet-benchmark`

[10]`http://www.github.com/elikip/bist-parser/`

[11]`https://github.com/UniversalDependencies/UD_English`