[Reviews · NeurIPS 2017]

Reviewer 1



The paper presents an automatic mechanism for converting unbatched computation graphs into batched computation graphs for efficient execution. The method works on dynamically specified graphs, and is therefore applicable to neural networks on strangely shaped data such as trees and graphs of varying topology. Since finding an ideal batching structure is NP hard, the paper proposes a few simple heuristics: these are not optimal but capture the bulk of the speedup due to batching. I am extremely happy with this paper. As someone who has worked with neural networks on tree shaped data, I can attest that batching is the source of most of the complexity. Static graph-based techniques such as TensorFlow Fold work but require significant refactoring of the code. This refactoring then complicates mundane issues, such as running out of memory when a batch of trees contains many large trees. My main request is a discussion of how best to deal with memory limits. The issue is that a batch of 32 trees requires a varying amount of memory depending on how large the trees are (the same applies to sequences, graphs, etc.). The easy approach is to try with a large batch and shrink if one runs out of memory, though the easiest way to do this changes the behavior of the estimator. Thus, different problems may warrant different ways of handling memory limits, and I am curious if the flexibility of the proposed scheme would hold up. Finally, a quantitative discussion of the cost of copying data into contiguous form would be interesting. Two specific statistics include (1) the fraction of elements that need copying and (2) the proportion of total time spent copying. In the worst case this copying could double the memory bandwidth; presumably the common case is much better, and I am curious how much.

Reviewer 2



Summary: The authors of this paper extend neural network toolkit DyNet with automatic operation batching. Batching enables efficient utilization of CPUs and GPUs by turning matrix-vector products into matrix-matrix products and reducing kernel launch overhead (for GPUs) but it is commonly done manually. Manual batching is manageable for simple feed-forward-networks but it becomes increasingly a headache as we explore more flexible models that take variable-length input, tree-structured input, or networks that perform dynamic control decisions. Chainer, DyNet, and PyTorch are recently proposed neural network toolkits that allow user to dynamically define the computation graph using the syntax of the host language (if, while, etc in python). This is desirable as it avoids tookit specific constructions (e.g., cond in TensorFlow) and make the network definition intuitive but it tends to limit performance because the network construction and computation happens at the same time. Thus although it would be straightforward to program a control flow that supports variable length or tree-structured input in these toolkits, in practice it would be too inefficient to process single instance at a time. The key contribution of this paper is to delay and automatically batch the computation so that a user can still define an arbitrary complex control flow using the host language as in original DyNet without worrying about operation batching. The approach is similar to TensorFlow Fold (TFF) but differs in two places: first, the computation graph is defined in a dynamic manner using the control flow of the host language and the user does not need to learn any new language (note that TFF introduces many new operations that are not in TensorFlow); second, it employs an agenda-based approach for operation batching in contrast to the depth-based approach employed in TFF. The authors empirically show that agenda-based batching is slightly more efficient than depth-based batching. Detailed comments: It wasn't clear to me if there is a stronger argument for dynamic definition of the computation graph other than the simplicity of coding compared to static definition of the graph in TFF, for example. Would the lazy evaluation still work when the control decision depends on the computed value (this is the case for the training with exploration scenario [2, 4, 9]. In this case, static graph definition may have an advantage.

Reviewer 3



This paper presents a solution to automatically do the operation batching in dynamic computation graphs. Their procedure contains three steps: graph definition, operation batching and determining execution order. They propose a ByAgenda batching algorithm and shows its superiority than the ByDepth algorithm in TensorFlow Fold. The problem is important and the solution sounds. One thing I am not sure about is the contribution other than the ByAgenda batching algorithm. Does TF Fold also utilizes the techniques in this paper about graph definition and determining execution order? I suspect the contribution of this paper is a little thin if it is all about a ByAgenda batching algorithm.